# Methods optimization for the expression and purification of human calcium calmodulin-dependent protein kinase II alpha

**Scott C. Bolton**[1,2], **David H. Thompson**[2], **Tamara L. Kinzer-Ursem**[1]*

**1** Weldon School of Biomedical Engineering, Purdue University, West Lafayette, Indiana, United States of America, **2** Department of Chemistry, Purdue University, West Lafayette, Indiana, United States of America

* tursem@purdue.edu

## Abstract

Calcium/calmodulin-dependent protein kinase II (CaMKII) is a complex multifunctional kinase that is highly expressed in central nervous tissues and plays a key regulatory role in the calcium signaling pathway. Despite over 30 years of recombinant expression and characterization studies, CaMKII continues to be investigated for its impact on signaling cooperativity and its ability to bind multiple substrates through its multimeric hub domain. Here we compare and optimize protocols for the generation of full-length wild-type human calcium/calmodulin-dependent protein kinase II alpha (CaMKIIα). Side-by-side comparison of expression and purification in both insect and bacterial systems shows that the insect expression method provides superior yields of the desired autoinhibited CaMKIIα holoenzymes. Utilizing baculovirus insect expression system tools, our results demonstrate a high yield method to produce homogenous, monodisperse CaMKII in its autoinhibited state suitable for biophysical analysis. Advantages and disadvantages of these two expression systems (baculovirus insect cell versus *Escherichia coli* expression) are discussed, as well as purification optimizations to maximize the enrichment of full-length CaMKII.

## Introduction

The CaMKII protein family is composed of four similar genes identified as α, β, γ, and δ [1]. While the isoforms γ, and δ are found ubiquitously in eukaryotic tissues, the α and β isoforms are found predominantly in the brain, making up 1–2% of total protein in the hippocampus, and play a role in the regulation of synaptic plasticity in neurons [2]. In the mammalian brain CaMKII is found as a heterogeneous oligomer, or holoenzyme, of twelve or fourteen subunits assembled from a mix of α and β isoforms [3–6]. Each individual subunit includes a kinase domain incorporating a regulatory segment, an oligomerization or hub domain, and a linker region joining the two domains whose length varies among isozymes. A number of works have focused on the structural assembly of CaMKII and its mechanism of activation [5–12]. It has been shown that the close proximity of neighboring subunits within the holoenzyme enables rapid inter-subunit cross-phosphorylation resulting in enzyme activation kinetics that are cooperative [10,13]. Transmission electron microscopy, crystallography, and small-angle x-ray

**Data Availability Statement:** All relevant data are within the paper and its Supporting information files.

**Funding:** The authors gratefully acknowledge the support of this work by the National Institute of Neurological Disorders and Stroke (NINDS) of the National Institutes of Health (NIH) under award number R21NS095218 (TKU) and the National Science Foundation (NSF) under Grant No. 1752366 (TKU). The funders had no role in study design, data collection and analysis, decision to publish, or preparation of the manuscript.

**Competing interests:** The authors have declared that no competing interests exist.

scattering have been used to reveal the morphology of subunit assembly and estimate the range of movement of the tethered kinase domains [5,6,8,9,14–17]. To reduce the complexity of understanding these multi-subunit mechanisms, these works have relied upon the recombinant expression of CaMKII where all subunits are of a single isozyme in the autoinhibited state. For high resolution structural studies, highly-enriched, autoinhibitied, monodisperse oligomeric CaMKII enzyme is needed. Further, like many other kinases, CaMKII is labile and shows decreasing activity and aggregation over time [18,19]. Thus, the conditions of purification and handling time must be optimized to ensure the recovery of the intact and functional CaMKII in high yield.

Many of the methods reported in the literature for generating purified recombinant CaMKII protein use bacterial expression or baculovirus insect expression, and these systems have employed non-tagged and polyhistidine tag fusion constructs. Table 1 lists the different purification strategies and yields (when given) that have been reported to date.

The Soderling lab first reported expression of recombinant wild type rat CaMKIIα using a baculovirus-insect expression vector system (BEVS). It was explained that cell homogenization into lysate in the presence of glycine betaine, a zwitterionic osmolyte that aids in protein stabilization and solubilization, prevented CaMKII from forming aggregates that would precipitate from solution during centrifugation [21,28]. The use of betaine in the lysis buffer resulted in improved recovery of CaMKII due to the added solubility, and this buffer is now commonly known as Brickey buffer. Using both ammonium sulfate precipitation and calmodulin-Sepharose (CaM-Sepharose) affinity chromatography purification steps, a final yield of 12–15 mg of purified CaMKII per liter of Sf9 cell culture was obtained, based an expression density of $3x10^6$ cells/mL. Török et al. reported a method omitting betaine, avoiding ammonium sulfate precipitation, and adding an anion exchange chromatography step [26]. Since betaine was not part of the protocol, the high yield might be due in part to the presence of 10% fetal bovine serum

**Table 1. Purification strategies for recombinant expression of full-length CaMKIIα.**

| Reference | Expression | Mutations | Step 1 (pH) | Step 2 (pH) | Yield (mg/L) |
|---|---|---|---|---|---|
| Waxham [20] | *E. coli* | – | CS (7.5) | – | – |
| Brickey [21] | BEVS | – | AS (7.5) | CS | 12–15 |
| Hagiwara [22] | *E. coli* | – | AS (7.6) | CS (7.5) | – |
| Putkey [23] | BEVS | – | AS (7.5) | CS** | – |
| Kolb [24] | BEVS<br>BEVS | –<br>N-terminal 6xHis<br>1–382, 1–427 | CS (7.5)**<br>IMAC (7.5) | S<br>– | –<br>– |
| Kolodziej [5] | BEVS | – | AS | IMAC | – |
| Singla [25] | BEVS | – | PC (7.0) | CS (7.3) | – |
| Török [26] | BEVS | – | CS (7.5) | Q (7.5) | 11 |
| Rosenberg [9] | BEVS | D135N | S (7.4) | Q (8.3)* | 0.005–0.01 |
| Chao [10] | *E. coli* | C-terminal 6xHis | IMAC | Q (8.3)* | 0.0025 |
| Coultrap [27] | HEK293T | – | PC (7.0) | CS | – |
| Myers [17] | BEVS | – | PC (7.2) | CS | – |

Legend: BEVS–baculovirus-insect expression vector system; AS–ammonium sulfate precipitation; CS–calmodulin-Sepharose affinity chromatography; S–cation exchange chromatography; Q–anion exchange chromatography; IMAC–Immobilized metal affinity chromatography; PC–phosphocellulose cation exchange chromatography;

*followed by a gel filtration step;

** followed by a sucrose density centrifugation step.

(FBS) added to the Sf9 growth medium that has been shown to limit non-specific binding of virus particles to cells and subsequently increase infection multiplicity [29].

The technique of Brickey et al. was adopted by the Waxham lab in a series of papers employing similar expression conditions and lysis buffers but differed in their purification methods. Initially, the purification steps were combinations of ammonium sulfate precipitation, CaM-Sepharose affinity chromatography, sucrose density gradient centrifugation, and phosphocellulose cation exchange chromatography [5,7,23,24]. These methods were eventually condensed to phosphocellulose cation exchange chromatography followed by CaM-Sepharose affinity chromatography [17,25,27].

In the Kuriyan lab, Rosenberg et al. used similar expression conditions as Török et al. (without mention of FBS supplementation) and modified the lysis buffer to replace betaine with 10% glycerol [9]. Although purification described a cation exchange chromatography step followed by an anion exchange chromatography step, an additional buffer exchange from HEPES buffer (for cation exchange) to Tris buffer (for anion exchange) was likely required and may have contributed to loss of CaMKII yield even before the final size-exclusion polishing step. Later, Chao et al. demonstrated the use of a bacterial system to express CaMKIIα [10]. Bacterial kinases can phosphorylate heterologous human kinase, including CaMKII, and to circumvent this problem the protein was co-expressed with λ-phosphatase [30]. The CaMKII gene also fused a C-terminal 6xHis tag to facilitate crude separation from lysate by use of an Ni-NTA affinity chromatography step, which would ideally capture full-length protein but not N-terminal truncated protein, though any C-terminal fragments due to full-length protein cleavage would also be captured. The three-step purification process included immobilized metal affinity chromatography (IMAC), anionic exchange chromatography and gel-filtration chromatography to produce 2.5 μg of purified CaMKII per liter of cells. Although this protocol has been used repeatedly, it requires large cultures [8,11,31–34].

Since the earliest reports of recombinant CaMKII expression over 30 years ago, studies of CaMKII continue to be highly relevant for studying kinase structure and function and $Ca^{2+}$-dependent cellular behavior [12,32–36]. The process of reconciling the many different purification methods using both expression systems motivated us to revisit the expression and purification of CaMKII. Here we document an optimized method to minimize handling time while maximizing yield of this labile enzyme.

## Materials and methods

### Baculovirus insect expression of CaMKII

**A pFastBac vector (GenScript) incorporating the wild type human CaMKIIα isoform 2 gene at cloning site BamHI-XhoI was transformed into chemically competent *E. coli* DH10BaC cells (Invitrogen) that contain a bacmid shuttle vector and helper plasmid to facilitate the transposition of the gene into baculovirus DNA [37]. Successful transformation was selected using TKG (Tetracycline, Kanamycin and Gentamicin) plates containing X-Gal (5-Bromo-4-Chloro-3-Indolyl β-D-Galactopyranoside, Invitrogen). White colonies were formed that indicated the disruption of the bacmid lacZα gene by successful transposition of the CaMKII gene from the donor pFastBac plasmid**. A single white colony was picked for bacmid prep and inoculated into LB medium containing TKG and incubated at 37˚C with shaking at 225 rpm for 16 h. The cells were pelleted and the baculovirus DNA extracted using a Qiagen mini-prep kit following the manufacturer's standard protocol. DNA was stored at 4˚C until use. ESF 921 growth medium (150 μL, Expression Systems) without antibiotics was mixed with 9 μL transfection reactant in a 24 well plate before addition of 1 μg baculovirus DNA and incubation at 20˚C for 30 min. Next, 850 μL of Sf9 cells ($2 \times 10^6$ cells/mL,

Expression Systems LLC, Davis, CA) were added to the well, sealed with a breathable membrane, and incubated for 5 h at 27˚C with 120 rpm shaking. Media (3 mL) containing 10% heat inactivated FBS (Sigma F4135) was added and the cells were incubated at 27˚C with 120 rpm shaking for 1 week. P0 viral stock was reserved from the supernatant of pellet centrifugation. Further viral amplifications were performed using adherent Sf9 cells in a T-75 flask containing ESF 921 growth medium plus 5% heat-inactivated FBS (Sigma) at 27˚C. After cells reached 50% confluence, 30 μL of viral stock was added and allowed to incubate 4–5 days. The amplified virus preparation was harvested from the media by centrifugation at 300 x g for 10 min, then passed through a sterile 0.22 μm filter (VWR). Viral amplification was iterated four times, creating P1, P2, P3, and P4 preparations. Lastly, protein expression was accomplished by incubating 2 mL of P4 stock (1% v/v) with 200 mL of Tni cells (Expression Systems LLC, Davis, CA) ($1x10^6$ cells/mL, unless otherwise stated) in ESF 921 growth medium in a 1 L spinner flask (Corning) at 27˚C with 140 rpm shaking, which resulted in a multiplicity of infection (MOI) of 3.2. After 24 h, the culture was supplemented with 10 mL (5%) Boost Production Additive (Expression Systems LLC, Davis, CA) where indicated. The culture was allowed to incubate with shaking for an additional 48 h (unless otherwise indicated). Cells were pelleted at 300 x g for 10 min, washed once with buffer (50 mM HEPES, pH 7.5, 200 mM NaCl, 1 mM EGTA and 1 mM EDTA) and pelleted again, then used immediately for purification or flash frozen in liquid $N_2$ and stored at -80˚C.

## Bacterial expression of CaMKII

A pD444-SR cDNA vector (ATUM, Newark, CA) containing a T5 promoter, a wild-type human CaMKIIα isoform 1 gene codon-optimized for E. coli expression, and a λ-phosphatase gene was transformed into chemically competent BL21-CodonPlus (DE3) (Agilent) cells following manufacture instructions and plated with ampicillin. Successful colonies were picked and amplified in 5 mL of LB media containing ampicillin overnight at 37˚C with shaking at 250 rpm. Protein was expressed by growing 200 μL of culture overnight in 200 mL of LB containing ampicillin at 37˚C with shaking at 250 rpm until $OD_{600}$ = 0.8. Cell cultures were chilled to 4˚C, induced with 0.5 mM isopropyl β-D-1-thiogalactopyranoside (IPTG) (Sigma) and 0.4 mM $MnCl_2$ and incubated at 16˚C with shaking at 250 rpm for 16 h (unless otherwise indicated). Cells were harvested by centrifugation at 3000 x g at 4˚C for 15 min, washed once with buffer (50 mM HEPES, pH 7.5, 200 mM NaCl, 1 mM EGTA and 1 mM EDTA), then flash frozen in liquid nitrogen and stored at -80˚C.

## Purification of CaMKII

Insect cells were processed based on modifications to the method described by Singla et al. [25] Briefly, frozen cells were thawed and re-suspended in ice-cold Brickey buffer (50 mM HEPES pH 7.2, 7.5 or 8.0, 5% betaine, 1 mM EGTA, 1 mM EDTA, and 1X HALT protease inhibitor (Thermo Scientific)). Insect cells were homogenized on ice with 10 strokes of a dounce homogenizer. Bacterial cells were incubated briefly with lysozyme on ice then lysed by probe sonication. After homogenization, crude lysate from either insect or bacterial culture was clarified by centrifugation at 12,000 x g for 20 min at 4˚C, followed by ultracentrifugation of the supernatant at 100,000 x g for 1 h at 4˚C. The supernatant was decanted and passed through a sterile 0.22 μm filter. The filtered sample was loaded at 0.5 mL/min onto an AKTA FPLC system fitted with either a Mono S 5/50 GL column or freshly prepared 5 mL fine mesh cellulose phosphate column (Sigma P/N C2258) equilibrated in binding buffer containing 50 mM HEPES (pH 7.2, 7.5 or 8.0), 100 mM NaCl, and 1 mM EGTA. The column was washed with 5 column volumes (CV) of binding buffer, then eluted using a 100 mM– 1 M NaCl

gradient in 4 CV. Peak fractions (typically 2–3 mL) were combined; 2 mM $CaCl_2$ and 10% glycerol were added and mixed. The sample was then transferred to a 5 mL tube containing 300 μL CaM-Sepharose resin (GE Healthcare) equilibrated in 50 mM HEPES (pH 7.2, 7.5 or 8.0), 2 mM $CaCl_2$, 200 mM NaCl, and 10% glycerol. The reaction mixture was incubated on a rotisserie for 1 h at 4˚C. The resin was washed 3 times (10 CV, followed by 6 CV, followed by another 6 CV) with wash buffer containing 50 mM HEPES (pH 7.5), 2 mM $CaCl_2$, 500 mM NaCl, and 10% glycerol. After each wash the resin was collected via centrifugation at 500 x g for 5 min. The resin was then incubated twice for 15 min, each using 1 CV of elution buffer containing 50 mM HEPES (pH 7.5), 400 mM NaCl, 4 mM EGTA and 30% glycerol. Aliquots of eluted protein were immediately flash frozen in liquid $N_2$ and stored at -80˚C. Protein concentration was measured by ultraviolet absorbance at 280 nm.

## Gel electrophoresis and Western blot

All SDS-PAGE gel experiments were performed using 4%– 20% gradient precast gels (BioRad Mini-PROTEAN) run at 170 V for 40 minutes in SDS Tris Glycine buffer. Gels were loaded with equal sample volume: clarified lysate gels contained 2 μL per lane and purification gels contained 5 μL per lane. Dual Ladder, 2X Laemmli buffer, equal volume loading. Blue Native PAGE gel experiments were performed in a cold room (4˚C) using 10% precast gels (BioRad Mini-PROTEAN) and run at 170 V for 55 minutes. Coomassie gels were stained with GelCode Blue Safe Stain (Thermo-Fisher, Waltham, MA) and imaged using a c600 gel imager (Azure Biosystems, Dublin, CA). Images are cropped and globally adjusted for contrast and brightness with care not to remove background noise.

Western blots were performed using a semi-dry apparatus (BioRad) to transfer proteins from the precast SDS-PAGE gel to a 0.45 μm polyvinylidene difluoride (PVDF) membrane. Transfers were run at 15 V for 30 minutes. Following transfer, PVDF membranes were incubated with Pierce protein-free blocking buffer (Thermo-Fisher, Waltham, MA) for 30 minutes under gentle rocking, followed by incubation with anti-CaMKII primary antibody 6G9 (Invitrogen MA1-048) diluted in buffer (50% Tris-buffered saline buffer with 0.1% Tween-20 (TBS-T) + 50% blocking buffer) overnight. Membranes were then washed 3X with TBS-T and incubated with secondary antibody (IRDye 680RD anti-mouse, Li-COR Biosciences GmbH) for 1 hour at room temperature with gentle rocking. Membranes were then washed 3X with TBS-T and imaged on a c600 gel imager (Azure Biosystems, Dublin, CA).

## CaMKII phosphorylation and activity

Phosphorylated CaMKII (Thr286P) was produced by a reaction previously described by Bradshaw et al. [38] Briefly, a mixture containing 18 μM subunit CaMKII, 50 μM CaM, 500 μM ATP, 500 μM $CaCl_2$ and 4 mM $MgCl_2$ was incubated on ice for 30 min. CaMKII phosphorylation was verified by Western blot staining with 22B1 anti-phospho-CaMKII antibody (Invitrogen MA1-047).

Kinase activity was determined by a radiolabeled ATP assay [18]. A reaction mixture containing 50 mM HEPES, pH 7.5, 200 μM $CaCl_2$, 10 mM $MgCl_2$, 1 mg/mL BSA, 1 μM CaM, and 100 μM Syntide-2 was incubated at 30˚C before adding 60 μCi/mL γ-$^{32}$P ATP in 100 μM cold ATP. The reaction was started by the addition of 10 nM CaMKIIα. Samples were spotted onto Whatman P81 paper every 15 s for the first minute, then every 30 s for the next two minutes. Papers were washed with 75 mM phosphoric acid four times and allowed to dry before scintillation counting.

### Negative stain TEM

A final polishing step to confirm isolation of 12-mer CaMKII holoenzymes was performed using a Superose 6 10/100 size-exclusion column resulting in an elution fraction of 20 μg/mL in final buffer 20 mM HEPES, pH 7.5, and 200 mM NaCl. The sample (3 μL) was then incubated on a glow-discharged, carbon-coated 400 mesh copper grid (CF400-Cu, Electron Microscopy Sciences) for 1 min, followed by washing with three drops of a freshly-prepared 1% uranyl formate solution, then wicked dry. Grids were imaged on an FEI Talos F200C equipped with a 4k x 4k BM-Ceta CCD camera operating at 200 kV in low-dose mode (10 e⁻/A²sec) at 73,000 X magnification and -5 μm defocus.

## Results

### Expression of CaMKII

CaMKII was expressed in both *E. coli* cells and Tni insect cells to compare and characterize the protein produced from both systems. For both systems, lysis buffers incorporated betaine to improve protein solubility, resulting in the clarified lyste containing the majority of the expressed CaMKII [21]. For the bacterial expression method, BL21(DE3) cells were transfected with two plasmids: one plasmid contained the CaMKIIα isoform A gene (Q9UQM7-1) codon-optimized for *E. coli* expression, and the second plasmid contained a Lambda-phosphatase gene used to dephosphorylate expressed CaMKII within the cytoplasm. It is known that CaMKII expression in *E. coli* generates a significant fraction of truncation products, resulting in a mixture of full-length subunits and kinase domain-truncated monomers [20,39]. To mitigate this problem, expression was carried out at a lower temperature of 16˚C. To observe the progression of CaMKII expression, 5 mL samples of expression culture were collected at various timepoints post-induction, and the samples were lysed, clarified, and analyzed in a Western blot stained with anti-CaMKII 6G9 that detects the kinase domain in both full-length and truncated subunits, and imaged with a near infrared secondary antibody (Fig 1A). At all the time points tested, a band at 50 kD indicating expression of full length CaMKII is seen, as is a second doublet band just below 37 kD that corresponds to the N-terminal truncated expression. All bands appear to be maximal at 18 h, full-length CaMKII expression does not appear to increase beyond 18 h, and the doublet band significantly decreases after 18 h. It was hypothesized that the doublet band below 37 kD may indicate the presence of a phosphorylated truncation product. However, staining with an anti-phospho-CaMKII antibody 22B1 revealed no presence of Thr286 phosphorylation (Fig 1B).

For CaMKII expression in BEVS, transfection and virus amplification was performed in adherent Sf9 cells, with amplification culture media supplemented with 5% FBS to allow for improved adherence, higher multiplicity of infection (MOI), and smaller volumes of intermediate titer stock. It was determined empirically that four iterations of 5-day amplification were sufficient to produce a virus concentration suitable for protein expression. Expression of CaMKII was performed by adding P4 virus to spinner cultures of Tni cells.

At various timepoints post-infection, samples of Tni cell culture were collected, lysed and clarified to perform Western blots with anti-CaMKII 6G9 as was done with bacterial samples. In Fig 1C, lanes 48 h– 96 h show a band at 50 kD expression of full-length CaMKII, a faint doublet band just above it, and a faint band at 30 kD. The bands appear strongest after 72 h post-infection and continued up to 96 h post-infection. Similarly to *E. coli* expression, a Western blot stained with a CaMKII phospho-antibody 22B1 showed that CaMKII was not phosphorylated at Thr286 (Fig 1D).

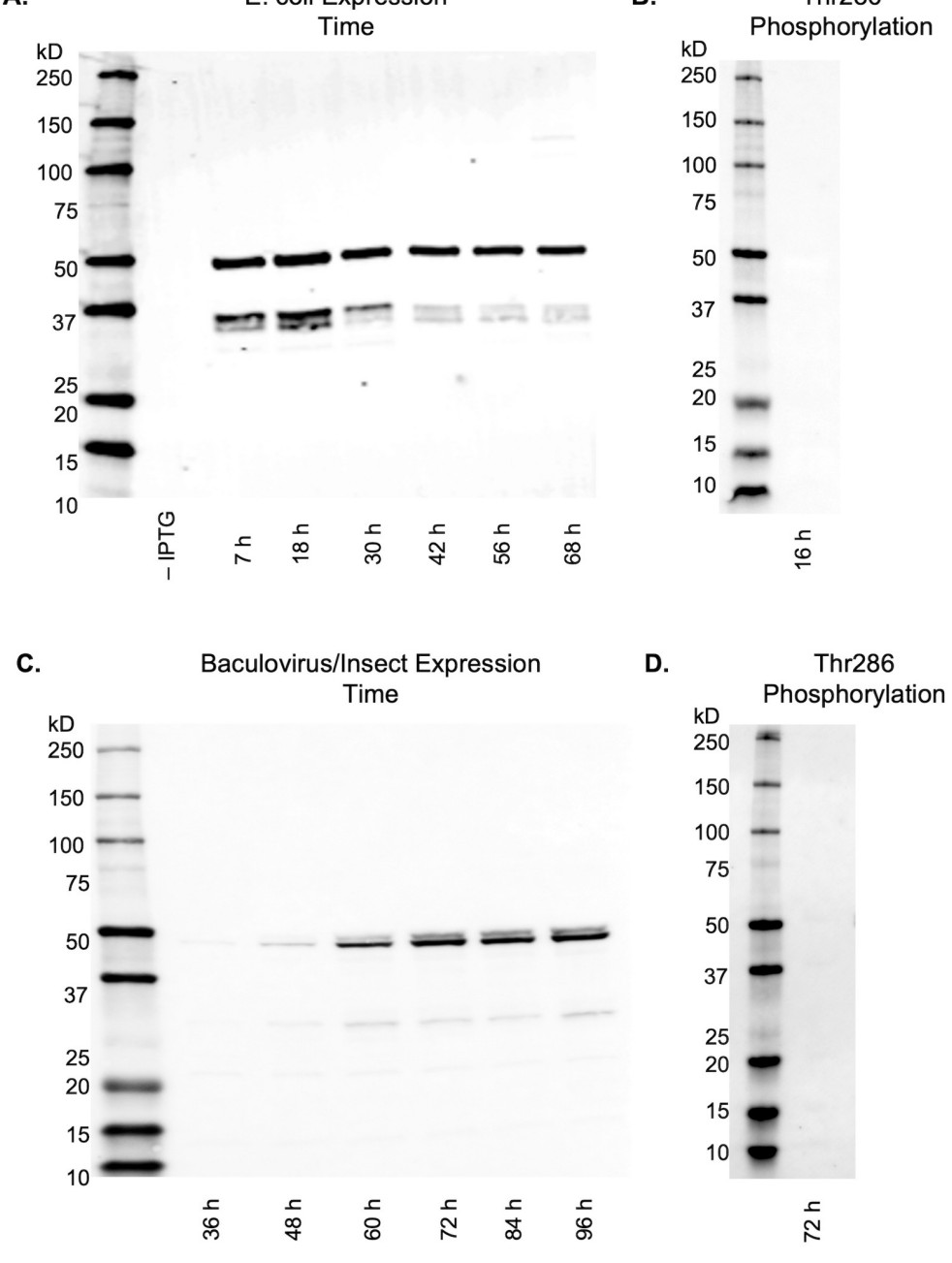

**Fig 1. Western blot analysis of CaMKII expression in clarified lysate as a function of time and expression system.** (A) *E. coli* expression at various timepoints detected with primary antibody anti-CaMKII 6G9 followed by IRDye 680RD. (B) *E. coli* expression (18 h) detected with primary antibody anti-phospho-CaMKII 22B1 followed by IRDye 680RD reveals no presence of Thr286 phosphorylation. (C) Tni expression at various timepoints detected with primary antibody anti-CaMKII 6G9 followed by IRDye 680RD. (D) Tni expression (72 h) detected with primary antibody anti-phospho-CaMKII 22B1 followed by IRDye 680RD reveals no presence of Thr286 phosphorylation.

BEVS expression results shows a doublet band at 50 kD, which is not found in the bacterial expression data. This is likely due to the small difference in sequences between the two expression systems. While the bacterial expression used CaMKIIα isoform Q9UQM7-1, BEVS expression used CaMKIIα isoform Q9UQM7-2 which encodes a 12-mer nuclear localization

sequence (NLS) containing the residues KKRKSSSSVQLM. This addition is found at the end of the linker region and has four consecutive serines that can be phosphorylated. Liquid chromatography–mass spectrometry (LC-MS-MS) analysis on BEVS CaMKII confirmed that the protein was not phosphorylated at Thr286 (S1 Table), Thr305 or Thr306 (S2 Table), but did show phosphorylation at the NLS serine residues in 53% of the fragment sequences (S3 Table).

Three additional parameters for the baculovirus/insect expression vector system were evaluated to determine the effect on protein yield: seed density, virus concentration, and the use of an expression additive to supplement the media (Fig 2). First, to evaluate the effect of seeding density on protein expression, a side-by-side expression was conducted where one flask had a starting cell count of $1 \times 10^6$ cells/mL and the other flask had a starting cell count of $2 \times 10^6$ cells/mL. Both flasks were infected with P4 virus at a MOI of 3.2 (1% v/v) and incubated for 72 h. Samples (5 mL) of lysate from both flasks were lysed, clarified, and analyzed in Western blots detected with primary antibody anti-CaMKII 6G9 followed by IRDye 680RD (Fig 2A). Both flasks appeared to produce a similar amount of full-length CaMKII, yet the expression seeded with $2 \times 10^6$ cells/mL showed increased truncation or degradation products.

Next, the effect of virus concentration on the expression of CaMKII was evaluated. Three identical flasks were each infected with virus at a MOI of 0.32 (0.1% v/v), MOI of 3.2 (1% v/v) or MOI of 16 (5% v/v). At 72 h post-infection, Western blots detected with primary antibody anti-CaMKII 6G9 followed by IRDye 680RD were performed on each culture (Fig 2B). The results demonstrate that the maximal expression of full-length CaMKII at 50 kD occurred when the virus concentration was 0.1%, however this was also accompanied by substantial

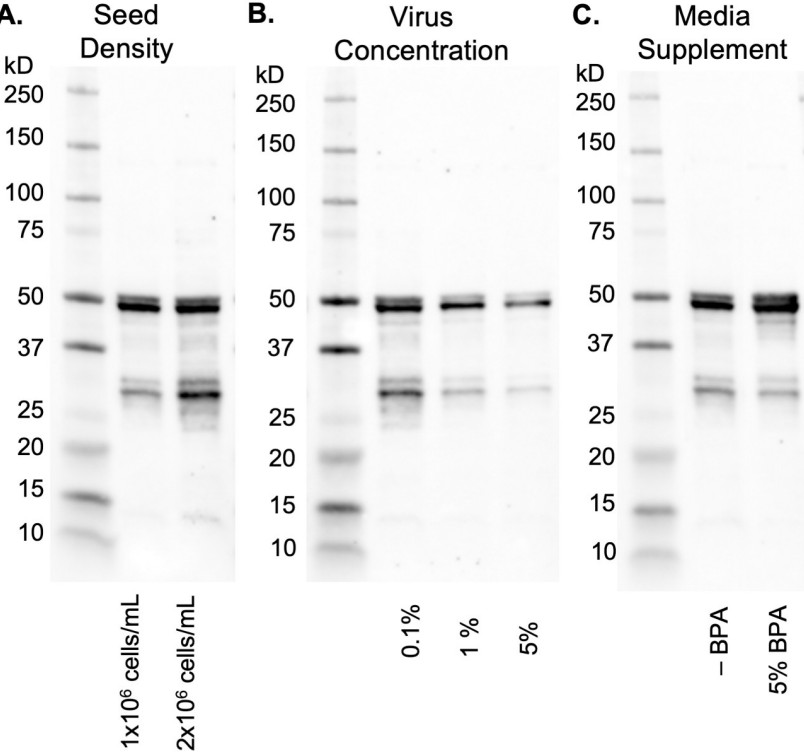

**Fig 2. Analysis of baculovirus/insect expression parameters on the expression of CaMKII.** All figures are Western blots stained with primary antibody anti-CaMKII 6G9 followed by IRDye 680RD. (A) Effect of seeding density of Tni cells before infection. (B) Effect of virus concentration at multiplicity of infections of 0.32, 3.2, and 16. (C) Effect of 5% BPA added 24 h post-infection (-BPA is no BPA added).

bands of CaMKII truncation or degradation products, especially one prominent 30 kD band. Higher virus concentrations of 1% and 5% showed less degradation, but also showed less full-length CaMKII formation, especially at 5% virus concentration.

Boost Production Additive (BPA) (Expression Systems LLC, Davis, CA) is a nutrient boost for cells during late-stage infection that may improve expression yield of some proteins. We examined the effect of using this additive on our expression experiments. Two 72 h expression cultures were initiated with a seed density of 1 x $10^6$ cells/mL and virus at a MOI of 3.2 (1% v/v), with one flask receiving 5% BPA after 24 h. Western blot stained with primary antibody anti-CaMKII 6G9 followed by IRDye 680RD showed that BPA enabled a greater amount of full-length CaMKII and a slightly lighter band of truncated products (Fig 2C).

## Purification of CaMKII

As discussed earlier and as shown in Table 1, many CaMKII purification protocols have used cation exchange chromatography in their methods, though at different stages in the purification scheme and with different buffer pH values [17,24,25,27,40]. Our motivation to use ion exchange chromatography as the first step was twofold: to rapidly isolate full-length CaMKII from lysate thus limiting potential protease activity, and to enrich CaMKII so that the subsequent calmodulin-Sepharose resin incubation volume would be small. Additionally, a rapid elution gradient was chosen to minimize both the run time and the elution volume. By fixing these parameters, we were able to evaluate electrostatic binding characteristics and CaMKII enrichment across a range of buffer pH values.

Three purifications at varying pH of CaMKII from Tni clarified lysate at were performed side-by-side. The low range of the pH value was bounded by both the ExPASy predicted iso-electric point of 6.6, and empirical data showing possible aggregation of CaMKII under ischemic conditions at or below pH 6.8 (though this also requires activation by $Ca^{2+}$/CaM) [41–44]. Thus, we chose pH values sampling a range between pH 7.2 and pH 8.0. The results were analyzed by comparing Western blot full-length CaMKII bands at 50 kD present in the elution fractions for each pH experiment (Fig 3A–3C) and (S3 Fig).

Purifications were made using frozen Tni cell pellets that were solubilized in Brickey buffer at the desired pH, homogenized using a dounce homogenizer, clarified by ultracentrifugation, passed through a 0.22 μm syringe filter, then immediately loaded onto a Mono S column. In order to verify CaMKII binding to the column during loading, we took samples during loading of the lysate onto the column (Fig 3A–3C, Flow through Peak lanes) and when the last of the flow through was washed from the column (Fig 3A–3C, Flow through Tail lanes). A rapid elution gradient from 100 mM NaCl to 1 M NaCl was used to elute CaMKII from the column (Fig 3A–3C, Elution 1–5 lanes).

The Western blot of CaMKII purified at pH 7.2 (Fig 3A) displayed a faint band at 50 kD in the flow through tail and no bands in the elution fractions 1–5 which indicated that CaMKII did not bind the column. In contrast, Western blots of purifications at pH 7.5 and pH 8.0 both demonstrate full-length CaMKII had bound to the column and was eluted from the column (Fig 3B and 3C, Elution 1–5 lanes). At pH 7.5, CaMKII was eluted primarily in lanes 2 and 3, while at pH 8.0 CaMKII was eluted in lanes 4 and 5, two column volumes later than pH 7.5. It should be noted that in all three blots in Fig 3, the clarified lysate, flow through peak, and flow through tail lanes had highly diluted CaMKII which reduced the visibility of the bands, while elution lanes 1–5 were enriched by a factor of 40 and had strongly visible bands. At pH 8.0 there were CaMKII degradation products bound to the column that eluted in tandem with the full-length protein.

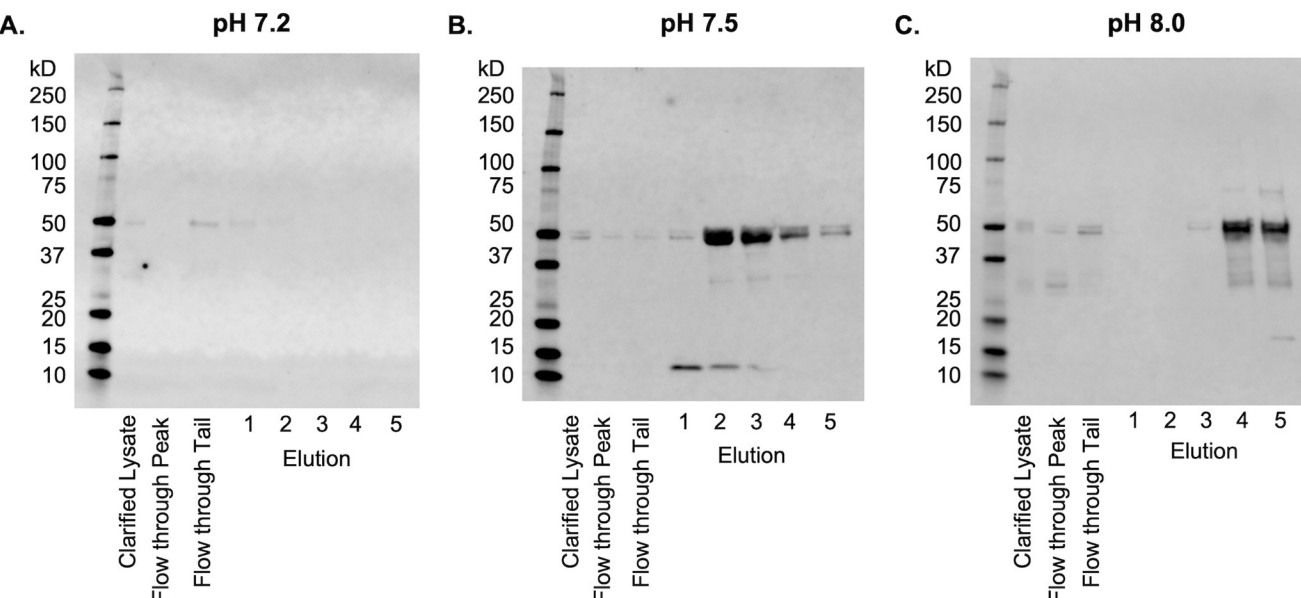

**Fig 3. Effect of pH on CaMKII purification with cation-exchange chromatography.** Initial separation of CaMKII from clarified lysate at (A) pH 7.2, (B) pH 7.5, and (C) pH 8.0. SDS-PAGE Western blot analysis detected with primary antibody anti-CaMKII 6G9 followed by IRDye 680RD.

Phosphocellulose resin has found use as a cation exchange chromatography medium in CaMKII purification protocols but has become increasingly difficult to purchase. The resin also requires extensive preparation including several rounds of fines removal, careful washing in acidic pH and basic pH buffers, then equilibration in buffer before pouring the column. In contrast, pre-packed ion-exchange columns are widely available and have robust handling characteristics including higher pressure and solvents for regeneration. We evaluated the performance of these two cation exchange resins by performing a side-by-side purification of CaMKII from clarified lysate with a 5 mL phosphocellulose column and a 1 mL Mono S column. Although cation exchange purification rapidly performs an initial separation of CaMKII from a large volume of lysate, non-specific protein debris in the eluted volume is still a significant problem. We employed a sequential CaM-Sepharose affinity chromatography step using a small (300 μL) fixed column volume in batch mode to further enrich CaMKII and remove cell debris.

Additionally, this second purification step also acts to normalize CaMKII enrichment between the two cation exchange chromatography columns (with differing elution volumes) and makes direct comparison a straightforward task. A schematic describing the side-by-side comparison is shown in Fig 4. Briefly, a previously frozen 5 g Tni pellet (corresponding to 200 mL of culture) containing expressed CaMKII was resuspended in Brickey lysis buffer, homogenized, clarified by ultracentrifugation, then divided into two volumes of equal size. One volume was loaded onto a freshly prepared phosphocellulose column and the other was loaded onto a Mono S column. Both columns used the same chromatography buffers and elution gradient, resulting in a 25 mL elution volume for Mono S and 125 mL elution volume for phosphocellulose (Fig 4). Each elution volume had 2 mM calcium and 10% glycerol added, followed by incubation with 300 μL of CaM-Sepharose resin on a rotisserie for one hour at 4 ˚C. After washing the resin, CaMKII was eluted with 4 mM EDTA. Since CaM-Sepharose column volumes were identical, elution volumes were also identical thus normalizing the final

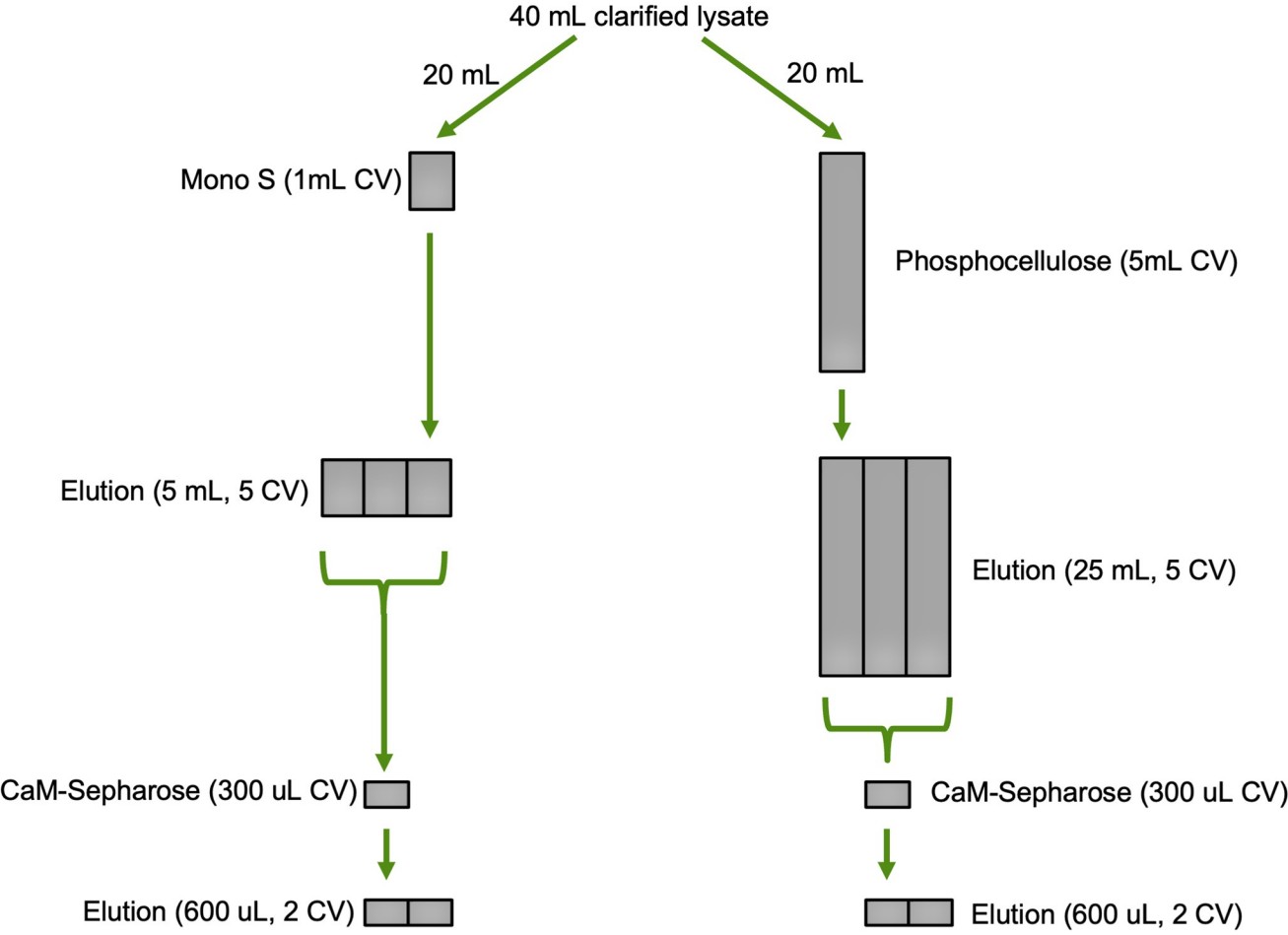

**Fig 4. Schematic detailing the comparison of the side-by-side two-step purification scheme.** The difference in column volume sizes between Mono S and phosphocellulose are normalized by the CaM-Sepharose step, thereby making protein concentration measurement between the two methods equivalent. Legend: CV–column volume.

concentrations. The final CaMKII yield from the method incorporating Mono S was 4.0 mg/L of culture, nearly twice the final yield of 2.2 mg/L of culture from the method using phosphocellulose (Fig 5).

One unexpected artifact observed in Fig 3C and 3D is the band of CaMKII present in wash step 2 in the Mono S purification that is not present in the phosphocellulose purification. This is attributed to some of the CaM-Sepharose resin accidentally being washed away when decanting the wash buffer from the resin after centrifugation. While this is not a common occurrence, we note that care must be taken when decanting after the wash and elution steps in batch reactions to maximize protein recovery. A representative set of Mono S / CaM-Sepharose blots that do not have this artifact is shown in S2 Fig.

## CaMKII yield comparison

In the previous sections we screened the growth and expression conditions in bacterial and insect systems to produce maximal full-length CaMKII in either system. We also optimized a two-stage purification system to enrich full-length CaMKII from cell lysate. With these steps

# Phosphocellulose followed by CaM-Sepharose

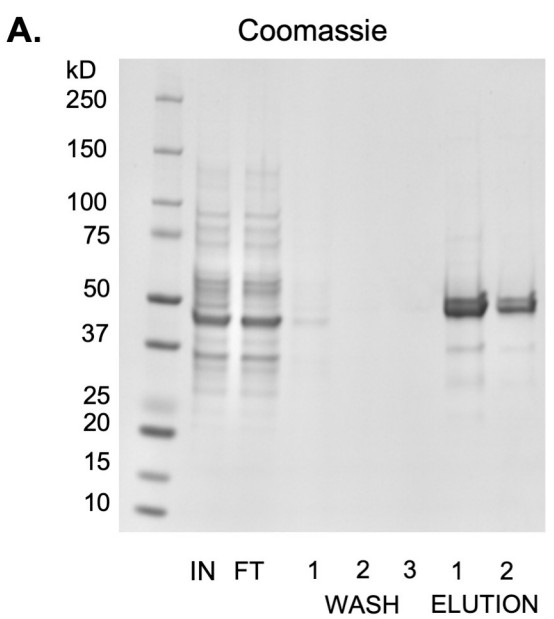

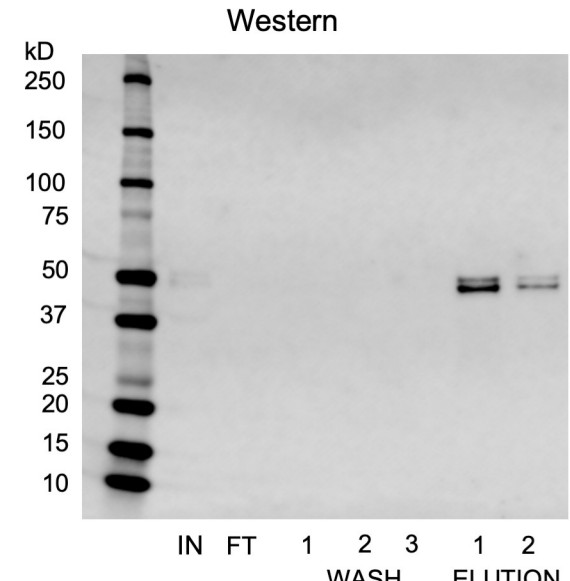

# Mono S followed by CaM-Sepharose

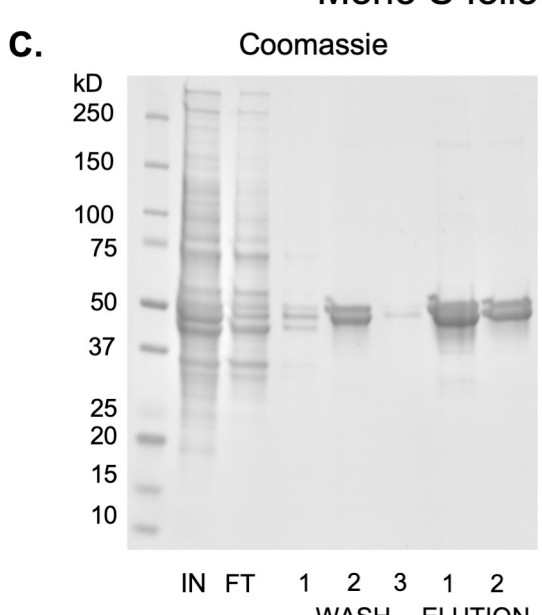

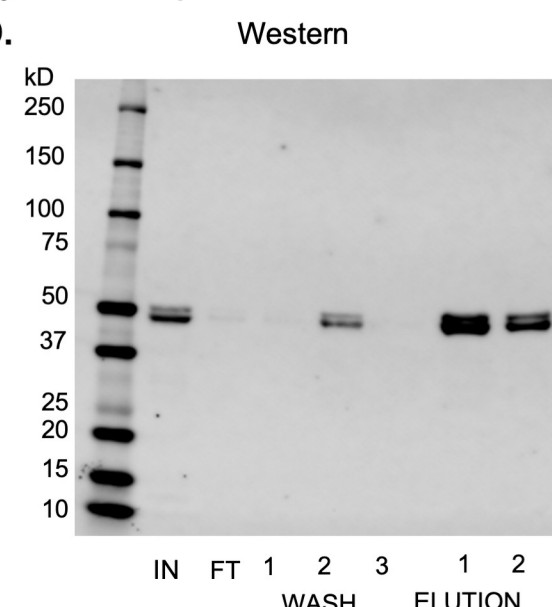

**Fig 5. Side-by-side comparison of CaMKII initial separation by either phosphocellulose or Mono S cation exchange chromatography, normalized by a second CaM-Sepharose affinity chromatography step.** SDS-PAGE followed by Coomassie stain (A and C) or Western blots stained with 6G9 anti-CaMKII primary antibody and detected with IRDye680RD secondary antibody (B and D). (A and B) use phosphocellulose for cation-exchange chromatography, and (C and D) use Mono S for cation-exchange chromatography. Lanes were loaded with equal volumes. Lane representation: IN–cation exchange chromatography combined elution fractions; FT–calmodulin-Sepharose flow through, WASH 1–3 – post-incubation wash steps, ELUTION 1–2 –elution column volumes.

optimized, we compared the yields of bacterial and BEVS CaMKII produced by this same two-stage chromatography method using bacterial and insect pellet masses of approximately 5 g each (Fig 6). Purification of CaMKII from the bacterial expression system produced approximately 400 μg of full-length CaMKII per liter of culture at a concentration of 0.45 mg/mL. This measurement, however, is compromised by the remaining impurity still found in the CaM-Sepharose elution fractions, specifically two bands at approximately 100 kDa and 30 kDa that were bound and eluted from the CaM-Sepharose resin (Fig 6A) but were not detected by anti-CaMKII antibody (Fig 6B). Purification of CaMKII from the BEVS system produced approximately 4 mg of full-length CaMKII per liter of insect culture at a concentration of 0.5 mg/mL.

## Structural characterization

The results shown in the SDS-PAGE gels in Figs 1–6 have demonstrated the production of monomeric full-length CaMKII as desired. Next, we characterized the assembly of recombinant CaMKII into oligomers using three different methods. First, a native polyacrylamide gel (7.5%) loaded with two-step purified CaMKII and stained with gel code blue showed a primary diffuse band just above the 720 kDa marker (expected mass is 648 kDa) (Fig 7A). A second faint band can be seen slightly higher which may represent the phosphorylation of the NLS. The second characterization method was to isolate CaMKII species using a Superose 6 size-exclusion column. Elution produced a predominant peak of dodecameric CaMKII at approximately 12 mL that coincides with a thyroglobulin standard peak at 660 kDa (Fig 7B) and also a minor flow through peak of higher masses that are likely to be aggregates.

Negative-stain TEM was used to characterize the morphology of CaMKII assembly into oligomers and confirm their monodisperse distribution on the grid. The TEM micrographs collected showed the characteristic assembly of CaMKII: a hub-and-spoke ring appearing in a face up orientation on the grid (Fig 7C). This morphology is consistent with prior observations where a circular hub domain with a center pore is decorated by punctate kinase domains randomly distributed within several nanometers of the hub circumference [5,6,8,17]. It is important to note that protein adsorption to the TEM grid and the negative staining procedure (blotting and sample dehydration) is expected to flatten and distort flexible oligomers such as CaMKII.

## Discussion and conclusions

Our motivation for this work is the exploration of the connected relationship between full-length protein expression in high yield and a purification strategy that maximizes enrichment while limiting the handling time to minimize protein degradation and aggregation.

Producing recombinant CaMKII in the lab remains a challenge. There are wide differences in the quality and quantity of CaMKII produced from bacterial systems and baculovirus insect expression systems. There are also large differences in purified protein yields obtained from the numerous purification schemes shown in the literature. Another consideration is the lab environment and the equipment available to produce and purify the protein. These combined factors significantly affect the yield of enriched, full-length dodecameric CaMKII available for experiments.

### Expression of CaMKII

Bacterial expression systems are ubiquitous in laboratories and are easier to set up and maintain than BEVS, and the cloning protocol is simpler than the multiple steps needed to produce baculovirus-infected insect cells. Bacterial cells incorporating the CaMKII gene are easily

## Bacterial Expression

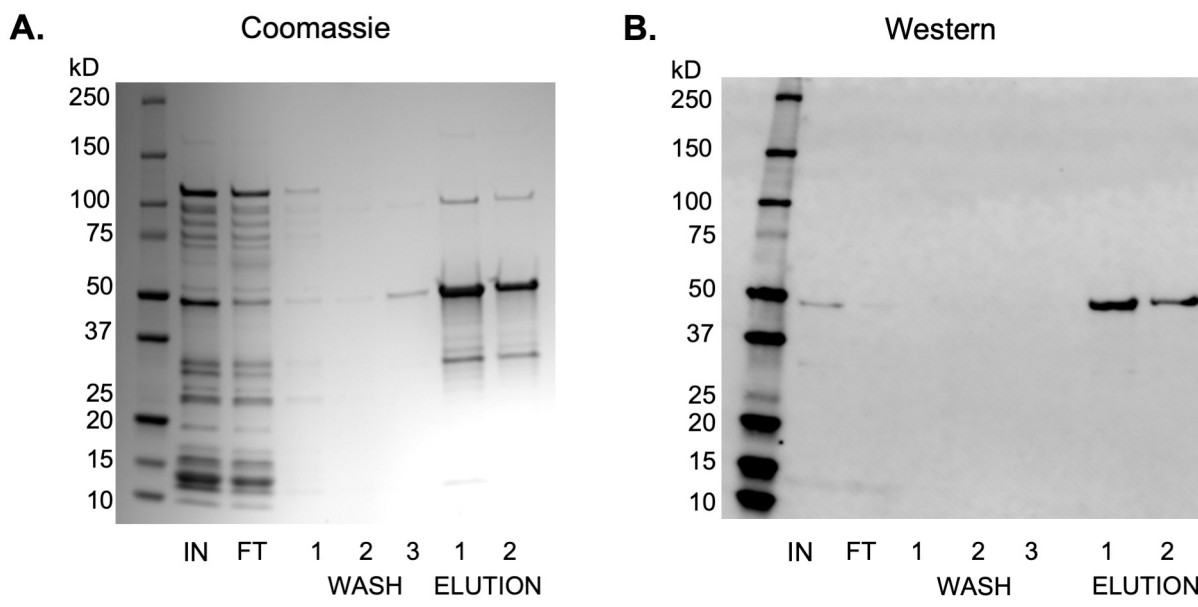

## BEVS Expression

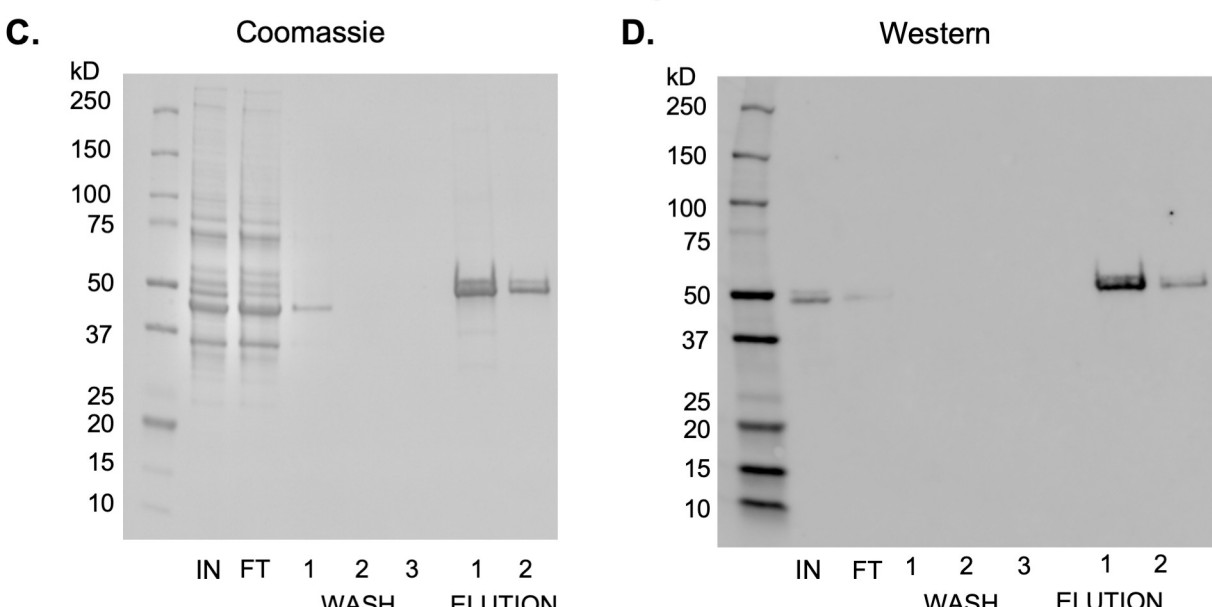

**Fig 6. Side-by-side CaMKII purification from bacteria and BEVS lysates.** The difference in column volume sizes between Mono S and phosphocellulose are normalized by the CaM-Sepharose step, thereby making the protein concentration measurement between the two methods equivalent. Legend: CV–column volume.

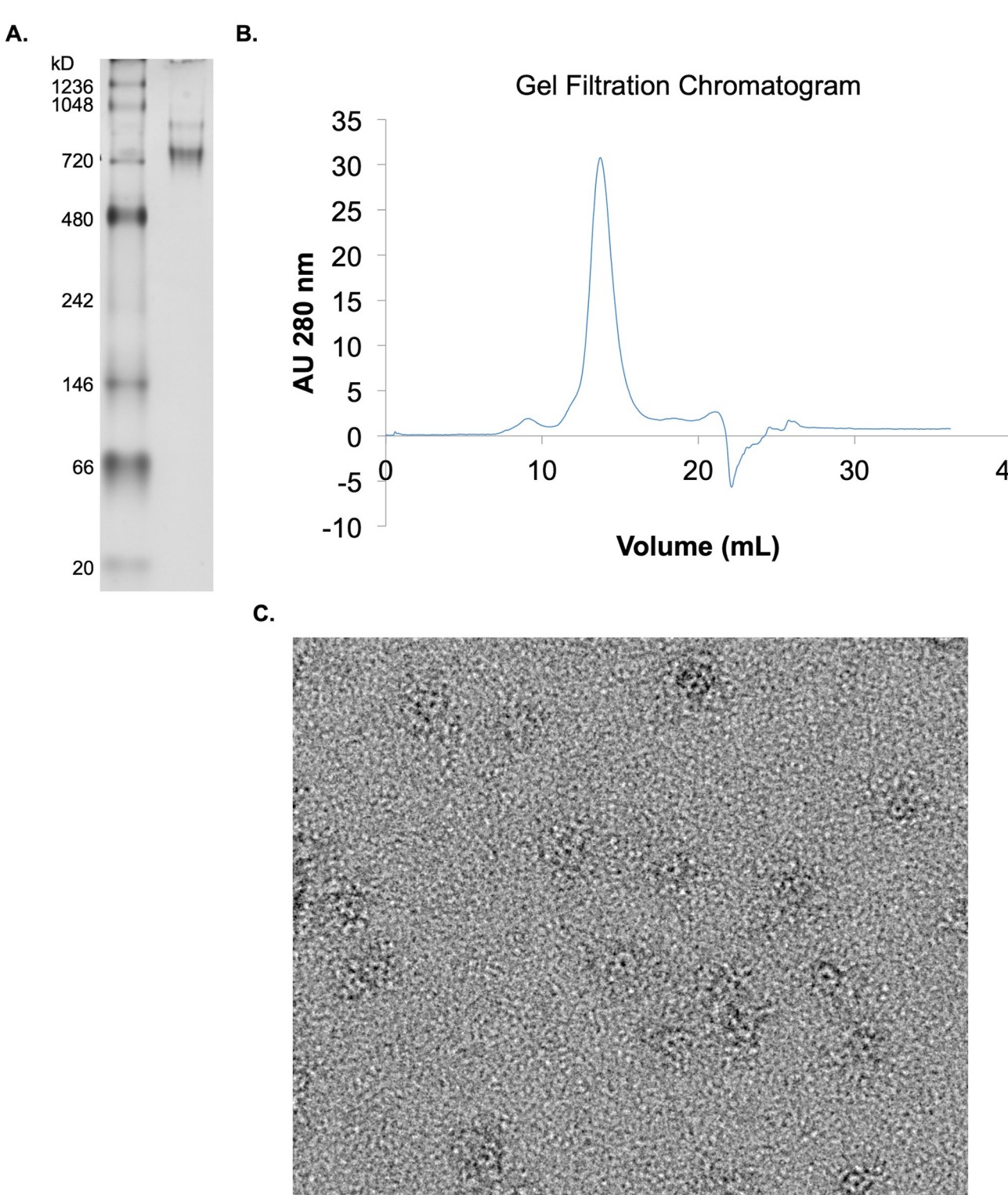

**Fig 7. Characterization of two-stage purified CaMKII obtained from BEVS.** (A) Native PAGE followed by Coomassie stain. (B) Superose 6 size-exclusion chromatogram. (C) Representative negative stain TEM micrograph of CaMKII oligomers. Scale bar 20 nm.

stored as glycerol stocks and readily scaled up to express protein as needed. Despite the ease of setup and use of bacterial systems, it is not always easier to produce full-length protein. Heterologous proteins may contain rare codons that result in early termination of translation that produces unwanted truncated products. Undesired phosphorylation of the target protein also requires effort to minimize. These difficulties may be ameliorated somewhat by the careful selection of expression temperature, induction rate, the use of codon optimization in the gene sequence, and the co-expression of phosphatases. Because the amount of full-length CaMKII produced per liter of bacterial culture is much less than with BEVS culture, far more culture volume–and resulting lysate volume–must be processed which is not time and cost efficient.

In contrast, insect cells require far more handling including regular passage, repeated growth and viability monitoring, and days of infection with virus to produce the target protein or make additional virus. Eukaryotic insect cells are, however, more friendly to mammalian kinases where codon usage is compatible, and genes may be cloned without codon substitutions. Phosphorylation and other post-translational modifications are a concern for BEVS, and for CaMKII, we found the NLS serine residues are phosphorylated during expression. Still, we found that the critical kinase-activation residues Thr286 and Thr305/Thr306 are not phosphorylated and do not require a de-phosphorylation step to yield autoinhibited (inactive) CaMKII when expressed in the BEVS system.

The first step compares protein expression quantity and contamination produced in clarified lysate between the best protocols of each expression system. Our method for measuring CaMKII is by Western blot using the monoclonal antibody 6G9, which recognizes the N-terminal kinase domain of CaMKII. Since both full-length and partially translated protein will contain some or all of the kinase domain, the 6G9 antibody is useful to detect even truncated protein species. We also use a second phospho-specific antibody to detect the subpopulation of CaMKII that is phosphorylated at Thr286.

In the bacterial expression system, a qualitative comparison between full-length and truncated protein detected over the expression time course shows that they are similar in amount in the first 24 hours, but the detection of these truncations declines substantially over longer time periods (Fig 1A), likely due to proteasome activity or potentially sequestration and excretion from the cells. Full-length CaMKII also appears to decrease after 18 hours, but not as markedly as the truncations. Thus, a longer expression time improves the relative purity of full-length protein over truncated protein. However, if a purification scheme can remove the truncations, then it is more beneficial to stop expression where full-length protein is maximized. Our conclusion is that a lower expression temperature, gentle induction, and expression for 18 h yields a maximal amount of full-length CaMKII provided that the large number of truncations can be removed efficiently.

In the BEVS, fewer truncations than full-length CaMKII are detected by Western blot, and these truncations do not appear to degrade over time. Thus, it is helpful to optimize the initial conditions to limit truncations and increase yield. These initial conditions include two interdependent factors: seeding density and infection concentration. The current study shows that an infection MOI of 3.2 and a culture at $1 \times 10^6$ cells/mL density allows for early-stage infection and cell doubling while still providing enough nutrients to produce protein in late-stage infection (Fig 2A). Infection at a higher seed density ($2 \times 10^6$ cells/mL) appears to increase degradation products without increasing full-length protein, likely because of the stress of cell density and increased nutrient consumption. Interestingly, the increased degradation products are also seen when a culture of $1 \times 10^6$ cells/mL density is infected with virus at a MOI of 0.32 (Fig 2B). The low infection concentration allows for more than one round of cell doubling and repeated virus amplification until late-stage infection is reached towards the end of cell viability in the media. However, an initial virus infection at a MOI of 16 produces a lower yield,

because cell replication is halted, and protein expression begins at or near the initial seeding density. Our conclusion is that seeding BEVS Tni cells at a density of $1 \times 10^6$ cells/mL infected at a MOI of 3.2 produced a maximal ratio of full-length of CaMKII to truncations when harvested at 72 h. Additionally, the use of BPA 24 h post-infection appears to produce an enhanced amount of full-length protein expression with no relative increase in truncations (Fig 2C).

It is important to note that the doublet band found at 50 kD in the BEVS Western blots is likely due to our CaMKII gene isoform B (Q9UQM7-2) that contains a short nuclear localization sequence (NLS). This addition inserts 12 residues (KKRKSSSSVQLM) into the linker region that connects the kinase and hub domains. Proteomic analysis reveals that while Thr286 and Thr305/THr306 are not phosphorylated during expression in BEVS, the four NLS serine residues (Ser332 –Ser335) are phosphorylated in 52% of the sample which can be seen as a distinct second band in SDS-PAGE. The phosphorylated protein will have slightly lower mobility due to the increase in negative charge [45].

## CaMKII purification

Having optimized full-length CaMKII expression in both systems, the next goal is to choose a purification strategy that maximizes enrichment while limiting the handling time to minimize protein degradation. Some publications report the use of CaMKII fused with polyhistidine tags to allow an IMAC crude separation from lysate, but this technique may present obstacles. Bacterially expressed protein often includes fragments from proteolysis and expression truncation [39]. If these fragments also contain the fused histag, it will cause co-elution contamination that must be removed in subsequent steps. The tags may be cleaved after expression, but this also requires further incubation time and purification steps which we wish to avoid. Lastly, IMAC columns are not highly selective for the target protein and will bind many non-specific proteins. The reduced flow rate needed to support the long residence time needed for histag binding, often 0.25 mL/min or less, translates to nearly 3 h of processing time in our experiment. The eluted protein must still be enriched further, and imidazole must be removed. Thus, we specifically chose to express and purify CaMKII without fusion tags.

Ammonium sulfate was used in the earliest publications and is very effective at protein enrichment of large volumes of lysate, but it leaves many other proteases and contaminants in the resulting pellet. It also requires the removal of the ammonium sulfate salt by another chromatographic step. Instead, the crude separation of clarified lysate using cation exchange proves to be an effective method of isolating CaMKII quickly. Starting with 40 mL of clarified lysate and a loading rate of 1 mL/min onto the column, we can bind CaMKII to the column within 40 min and elute enriched protein with 5 column volumes (5 mL) when using a Mono S column at pH 7.5. The smaller cation elution volume is advantageous when performing the next enrichment step.

An interesting phenomenon is observed with cation exchange chromatography. Earlier publications report cationic exchange purification buffers between pH 7–7.5. The theoretical isoelectric point for a full-length CaMKII subunit is approximately 6.8, which should make it negatively charged above pH 7 and theoretically not bind cation resin. In our work, we show that CaMKII very weakly binds cation resin at pH 7.2 (Fig 3A), which is congruent with a small net charge resulting from a single, globular volume. As the pH increases, we would expect there to be no binding to the cation resin. Surprisingly, the opposite occurs.

An explanation for this disparity may be that as the pH increases, the quaternary shape and exposed surfaces of CaMKII may change. If we consider the catalytic and association domains of CaMKII independently, the kinase domain has a theoretical pI of 8.05, and the association

domain has a theoretical pI of 5.71. In order for CaMKII to bind strongly to the cation resin at pH 8, the kinase domains may be in a state of extension away from the hub, revealing positively charged surfaces that readily bind the cation resin. As seen in the results, a pH of 7.5 gave the best balance between good binding and efficient elution into a minimal volume of buffer for subsequent steps (Fig 3B).

The comparison of the Coomassie stained gels and Western blot analyses of the freshly prepared phosphocellulose column and the commercial Mono S column show similar amounts of full-length CaMKII and other non-specific debris. Therefore, we conclude that the preparation time saved and long-term resin stability of Mono S make it a preferable choice for the cation exchange chromatographic step.

Purification of CaMKII with CaM-Sepharose is widely used and can be done with either plug flow elution in EGTA buffer (on an FPLC) or using a batch incubation method, as we perform here. Batch mode has two advantages for our purification. First, we can use a much smaller volume of CaM-Sepharose resin slurry in a batch incubation than the typical 1 mL or 5 mL volume FPLC column. Thereby reducing the elution volume and increasing our enrichment. Second, it is also straightforward to include glycerol, needed for cryo-preservation, in wash and elution buffers. With an FPLC, however, adding glycerol to buffers increases the pressure against the column and often requires reducing and monitoring method flow rates to run below the column pressure limit.

In side-by-side purifications of CaMKII from BEVS and bacterial expression using the Mono S / CaM-Sepharose purification protocol, BEVS yielded ten-fold more full-length CaMKII per liter of culture than bacterial expression (an average of 4 mg at 0.5 mg/mL versus 0.4 mg at 0.45 mg/mL, respectively).

In comparing our BEVS yield to earlier work, our method produced less protein than reported by Brickey et al. Possible reasons for this discrepancy are reduced expression when using our viral construct in Tni cells, the loss of CaMKII in the cation exchange chromatography flow through, and limiting the elution volume of that step to 5 column volumes in order to minimize the CaM-Sepharose purification batch volume. Specific activity of our protein (5 μmol/min/mg) was consistent with earlier reports and showed a linear response ($R^2$ = 0.9989) for at least 90 seconds [21,25] (S4 Fig) When we embarked on the production of purified, recombinant CaMKII for our lab, the methods reported in the literature were diverse and the results varied greatly. It was not clear how to best select the parameters of expression system type, culture volume, pH, ion-exchange chromatography type, and/or affinity chromatography method in such a way that minimizes effort and handling time while demonstrating enriched yield of purified, mondisperse, oligomeric CaMKII. The advantage of this work is to show reduced preparation and handling time by choosing the best parameters that minimize the volumes of expression culture and the volumes of chromatographic columns. We do this by combining a smaller (200 mL) volume of baculovirus-insect culture with a pre-prepared, small 1 mL column volume Mono S ion-exchange chromatography method, targeted at pH 7.5, followed by batch-method Cam-Sepharose affinity chromatography. We hope that this detailed method optimization and comparison of methods provides clarity to the field for the efficient and reproducible generation of full-length, recombinant CaMKII protein.

## Supporting information

**S1 Fig. Mono S chromatograms of CaMKII crude separation at three different working pH values.** (A) pH 7.2, (B) pH 7.5, (C) pH 8.0.
(TIF)

**S2 Fig. Representative SDS-PAGE Coomassie gel and 6G9 Western blot of the two-step Mono S / CaM-Sepharose purification.** (A). SDS-PAGE followed by Coomassie stain, (B) Western blots stained with 6G9 anti-CaMKII primary antibody and detected with IRDye680RD secondary antibody.
(TIF)

**S3 Fig. Effect of pH on CaMKII purification with cation-exchange chromatography.** Initial separation of CaMKII from clarified lysate at (A) pH 7.2, (B) pH 7.5, and (C) pH 8.0. SDS-PAGE followed by Coomassie stain directly corresponds to the Western blots in Fig 3.
(TIF)

**S4 Fig. CaMKII specific activity after purification.** Radiolabeled ATP assay shows a linear response for at least 90 seconds. Specific activity is 5 μmol/min/mg. Error bars represent n = 3.
(TIF)

**S1 Table. Liquid chromatography mass spectrometry detection of CaMKIIα isoform B phosphorylation at Thr 286.**
(DOCX)

**S2 Table. Liquid chromatography mass spectrometry detection of CaMKIIα isoform B phosphorylation at Thr 305 / Thr 306.**
(DOCX)

**S3 Table. Liquid chromatography mass spectrometry detection of CaMKIIα isoform B phosphorylation at the nuclear localization sequence (Ser 332/333/334/335).**
(DOCX)

## Acknowledgments

The authors thank Prof. Andy Hudmon (Department of Medicinal Chemistry and Molecular Pharmacology, Purdue University) for his help in phosphocellulose preparation, and Dr. Keith Viccaro (Department of Medicinal Chemistry and Molecular Pharmacology, Purdue University) for his help in collecting kinase activity data. We also thank the Purdue Proteomics Facility for assistance with mass spectrometry data collection. We thank members of the Kinzer-Ursem lab for their helpful comments on the manuscript.

## Author Contributions

**Conceptualization:** Scott C. Bolton, David H. Thompson, Tamara L. Kinzer-Ursem.

**Data curation:** Scott C. Bolton.

**Formal analysis:** Scott C. Bolton.

**Funding acquisition:** Tamara L. Kinzer-Ursem.

**Investigation:** Scott C. Bolton.

**Methodology:** Scott C. Bolton.

**Project administration:** Tamara L. Kinzer-Ursem.

**Resources:** Tamara L. Kinzer-Ursem.

**Supervision:** David H. Thompson, Tamara L. Kinzer-Ursem.

**Validation:** Scott C. Bolton.

**Visualization:** Scott C. Bolton.

**Writing – original draft:** Scott C. Bolton.

**Writing – review & editing:** Scott C. Bolton, David H. Thompson, Tamara L. Kinzer-Ursem.

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
