## [Decision Letter · Decision Letter 0]

30 May 2023

PONE-D-23-12570Methods Optimization for the Expression and Purification of Human Calcium Calmodulin-Dependent Protein Kinase II AlphaPLOS ONE

Dear Dr. Kinzer-Ursem,

Thank you for submitting your manuscript to PLOS ONE. After careful consideration, we feel that it has merit but does not fully meet PLOS ONE’s publication criteria as it currently stands. Therefore, we invite you to submit a revised version of the manuscript that addresses the points raised during the review process.

We look forward to receiving your revised manuscript.

Kind regards,

Jian Xu, Ph.D.

Academic Editor

PLOS ONE

2. PLOS ONE now requires that authors provide the original uncropped and unadjusted images underlying all blot or gel results reported in a submission’s figures or Supporting Information files. This policy and the journal’s other requirements for blot/gel reporting and figure preparation are described in detail at https://journals.plos.org/plosone/s/figures#loc-blot-and-gel-reporting-requirements and https://journals.plos.org/plosone/s/figures#loc-preparing-figures-from-image-files.

When you submit your revised manuscript, please ensure that your figures adhere fully to these guidelines and provide the original underlying images for all blot or gel data reported in your submission. See the following link for instructions on providing the original image data: https://journals.plos.org/plosone/s/figures#loc-original-images-for-blots-and-gels.

Reviewers' comments:

Reviewer's Responses to Questions

**Comments to the Author**

1. Is the manuscript technically sound, and do the data support the conclusions?

Reviewer #1: Partly

Reviewer #2: Yes

Reviewer #3: Partly

2. Has the statistical analysis been performed appropriately and rigorously? 

Reviewer #1: N/A

Reviewer #2: N/A

Reviewer #3: N/A

3. Have the authors made all data underlying the findings in their manuscript fully available?

Reviewer #1: Yes

Reviewer #2: Yes

Reviewer #3: Yes

4. Is the manuscript presented in an intelligible fashion and written in standard English?

Reviewer #1: Yes

Reviewer #2: Yes

Reviewer #3: Yes

5. Review Comments to the Author

Reviewer #1: This paper compares two systems (insect cell vs bacteria) for the expression and purification of Human Calcium

Calmodulin-Dependent Protein Kinase II Alpha. The paper is generally very well presented in terms of readability, logical ordering, and presentation and discussion of data. I am sure it will be useful for others trying to express similar complex proteins. However, for it to be most useful to others, the following additional information should be supplied that relates to the baculovirus expession aspect:

Source of Sf cells (line 91)

Source of DH10Bac (line 86) and brief explanation of what these cells are or reference to a method for making recombinant baculoviruses using the 'Bac-to-Bac' method.

The titre of the recombinant virus produced and the titre/MOI used to infect cells. The authors mention the importance of MOI (line 189) but do not mention when they describe infection details (lines 190-191 and in Fig 1). The infection details are too vague.

In Fig 1 Panel C there is no negative control for noninfected or mock-infected cells.

Reviewer #2: The manuscript by Bolton et al. optimized an expression and purification method of Human CaMKIIα. Obtaining enough full-length protein is of great significance for the functional study of CaMKIIα, and this study provides a feasible solution. Therefore, I recommend publication in PlosONE after minor modifications.

(1)Line 99 and 219, I suggest converting the volume ratio to MOI (multiplicity of infection). For the baculovirus insect expression system, the MOI is a useful parameter, whereas the volume ratio does not provide sufficient information for the reader to repeat this study. For this, the titer of the P4 viral stock needs to be determined.

(2)Line 192 and 195, "post-infection" instead of "post-induction".

Reviewer #3: Comments:

The manuscript "Methods optimization for the Expression and Purification of Human Calcium Calmoudulin-Dependent Protein Kinase II Alpha" describes studies aimed at applying E. coli and baculovirus protein expression technology and accompanying purification schemes to develop an optimized process for obtaining high yields of CaMKIIalpha.

1. The authors did not quantify the amount of recombinant virus used in any of the studies. The authors use a percentage of virus for their studies. For completeness the authors should have quantified the virus titer of their P4 preparation.

2. Why did the authors choose Tni vs. Sf9 cells for their 200 ml production runs?

3. The authors did not clearly state the advantages of their "optimized" methodology over that of the previously published studies using BEVS.

4. I did not see ref. 8-10 mentioned within the manuscript text. Ref 10 is cited in Table 1.

5. The quantity of protein loaded onto gels was not stated.

6. Ref 2 is not in the text. I suspect it should be linked with ref. 1.

7. Ref 7- protein kinase II1 1. I believe 1 1 should be deleted.

6. PLOS authors have the option to publish the peer review history of their article (what does this mean?). If published, this will include your full peer review and any attached files.

Reviewer #1: No

Reviewer #2: No

Reviewer #3: No

---

## [Author Response · Author response to Decision Letter 0]

8 Jul 2023

PONE-D-23-12570: Methods Optimization for the Expression and Purification of Human Calcium Calmodulin-Dependent Protein Kinase II Alpha

Response to Reviewers

Reviewer #1: 

This paper compares two systems (insect cell vs bacteria) for the expression and purification of Human Calcium

Calmodulin-Dependent Protein Kinase II Alpha. The paper is generally very well presented in terms of readability, logical ordering, and presentation and discussion of data. I am sure it will be useful for others trying to express similar complex proteins. However, for it to be most useful to others, the following additional information should be supplied that relates to the baculovirus expression aspect:

(1) Source of Sf cells (line 91)

Expression Systems has been added as the source on line 91.

(2) Source of DH10Bac (line 86) and brief explanation of what these cells are or reference to a method for making recombinant baculoviruses using the 'Bac-to-Bac' method.

Invitrogen has been added as the source on line 89. 

A pFastBac vector (GenScript) incorporating the wild type human CaMKIIɑ isoform 2 gene at cloning site BamHI-Xhol was transformed into chemically competent E. coli DH10BaC cells (Invitrogen) that contain a bacmid shuttle vector and helper plasmid to facilitate the transposition of the gene into baculovirus DNA (PMID 21390847). Successful transformation was selected using TKG (Tetracycline, Kanamycin and Gentamicin) plates containing X-Gal (5-Bromo-4-Chloro-3-Indolyl β-D-Galactopyranoside, Invitrogen). White colonies were formed that indicated the disruption of the bacmid lacZ� gene by successful transposition of the CaMKII gene from the donor pFastBac plasmid. This clarification has been added on line 87 - 93 in the Methods section.

(3) The titre of the recombinant virus produced and the titre/MOI used to infect cells. The authors mention the importance of MOI (line 189) but do not mention when they describe infection details (lines 190-191 and in Fig 1). The infection details are too vague.

The manuscript has been updated to include MOI values where relevant throughout the document including figure 2.

(4) In Fig 1 Panel C there is no negative control for noninfected or mock-infected cells.

This is true. Instead in Fig 1 at the first timepoint (36 hours post infection) we repeatedly found that very little protein was produced. Thus this is shown on the gel instead of the true negative control. The overall conclusion that we draw from the expression time course study remains the same. 

Reviewer #2: 

The manuscript by Bolton et al. optimized an expression and purification method of Human CaMKIIα. Obtaining enough full-length protein is of great significance for the functional study of CaMKIIα, and this study provides a feasible solution. Therefore, I recommend publication in PlosONE after minor modifications.

(1) Line 99 and 219, I suggest converting the volume ratio to MOI (multiplicity of infection). For the baculovirus insect expression system, the MOI is a useful parameter, whereas the volume ratio does not provide sufficient information for the reader to repeat this study. For this, the titer of the P4 viral stock needs to be determined.

The manuscript has been updated to include MOI values where relevant throughout the document including figure 2.

(2 )Line 192 and 195, "post-infection" instead of "post-induction".

The manuscript has been changed to “post-infection”.

Reviewer #3: Comments:

The manuscript "Methods optimization for the Expression and Purification of Human Calcium Calmoudulin-Dependent Protein Kinase II Alpha" describes studies aimed at applying E. coli and baculovirus protein expression technology and accompanying purification schemes to develop an optimized process for obtaining high yields of CaMKIIalpha.

1. The authors did not quantify the amount of recombinant virus used in any of the studies. The authors use a percentage of virus for their studies. For completeness the authors should have quantified the virus titer of their P4 preparation.

The manuscript has been updated to include MOI values where relevant throughout the document including figure 2.

2. Why did the authors choose Tni vs. Sf9 cells for their 200 ml production runs?

We chose Tni cells for protein expression because it had been recommended by the vendor (Expression Systems) for greater protein expression over Sf9/Sf21.

3. The authors did not clearly state the advantages of their "optimized" methodology over that of the previously published studies using BEVS.

Thank you for pointing this out. The previously published studies use parts of these methods in differing combinations. However, it is not clear from any single publication how to best select the parameters of expression system type, culture volume, pH, ion-exchange chromatography type, and/or affinity chromatography method in such a way that minimizes effort and handling time while demonstrating enriched yield of purified, mondisperse, oligomeric CaMKII. Our advantage is that we show reduced preparation and handling time by choosing the best parameters that minimize our volumes of culture and chromatographic columns. We do this by combining a smaller (200 mL) volume of baculovirus-insect culture with a pre-prepared, small 1 mL column volume Mono S ion-exchange chromatography method, targeted at pH 7.5, followed by batch-method Cam-Sepharose affinity chromatography. This has now ben added to the discussion.

4. I did not see ref. 8-10 mentioned within the manuscript text. Ref 10 is cited in Table 1.

References 8-10 were removed from an earlier text revision that included beta-CaMKII, The references accidentally remained in the bibliography (and Table 1) after editing. This has been fixed.

5. The quantity of protein loaded onto gels was not stated.

For protein purification gels, lanes were loaded with equal volumes (2�l for gels with cell lysate, and 5�l for gels with purification steps). The quantity of protein for each lane was not measured. This explanation has now been added to the methods section (line 148 – 149).

6. Ref 2 is not in the text. I suspect it should be linked with ref. 1.

The references have been fixed. 

7. Ref 7- protein kinase II1 1. I believe 1 1 should be deleted.

The references have been fixed.

---

## [Decision Letter · Decision Letter 1]

19 Jul 2023

Methods Optimization for the Expression and Purification of Human Calcium Calmodulin-Dependent Protein Kinase II Alpha

PONE-D-23-12570R1

Dear Dr. Kinzer-Ursem,

We’re pleased to inform you that your manuscript has been judged scientifically suitable for publication and will be formally accepted for publication once it meets all outstanding technical requirements.

Kind regards,

Jian Xu, Ph.D.

Academic Editor

PLOS ONE

Additional Editor Comments (optional):

Reviewers' comments:

Reviewer's Responses to Questions

**Comments to the Author**

1. If the authors have adequately addressed your comments raised in a previous round of review and you feel that this manuscript is now acceptable for publication, you may indicate that here to bypass the “Comments to the Author” section, enter your conflict of interest statement in the “Confidential to Editor” section, and submit your "Accept" recommendation.

Reviewer #1: All comments have been addressed

Reviewer #2: All comments have been addressed

Reviewer #3: All comments have been addressed

2. Is the manuscript technically sound, and do the data support the conclusions?

Reviewer #1: Yes

Reviewer #2: Yes

Reviewer #3: Yes

3. Has the statistical analysis been performed appropriately and rigorously? 

Reviewer #1: N/A

Reviewer #2: N/A

Reviewer #3: N/A

4. Have the authors made all data underlying the findings in their manuscript fully available?

Reviewer #1: Yes

Reviewer #2: Yes

Reviewer #3: Yes

5. Is the manuscript presented in an intelligible fashion and written in standard English?

Reviewer #1: Yes

Reviewer #2: Yes

Reviewer #3: Yes

6. Review Comments to the Author

Reviewer #1: (No Response)

Reviewer #2: (No Response)

Reviewer #3: (No Response)

7. PLOS authors have the option to publish the peer review history of their article (what does this mean?). If published, this will include your full peer review and any attached files.

Reviewer #1: No

Reviewer #2: No

Reviewer #3: No

---

## [Editor Report · Acceptance letter]

9 Aug 2023

PONE-D-23-12570R1 

Methods Optimization for the Expression and Purification of Human Calcium Calmodulin-Dependent Protein Kinase II Alpha 

Dear Dr. Kinzer-Ursem:

I'm pleased to inform you that your manuscript has been deemed suitable for publication in PLOS ONE. Congratulations! Your manuscript is now with our production department. 

Kind regards, 

on behalf of

Dr. Jian Xu 

Academic Editor

PLOS ONE